# Differential Regulation of Ca^2+^-Activated Cl^−^ Channel TMEM16A Splice Variants by Membrane PI(4,5)P_2_

**DOI:** 10.3390/ijms22084088

**Published:** 2021-04-15

**Authors:** Woori Ko, Byung-Chang Suh

**Affiliations:** Department of Brain and Cognitive Sciences, Daegu Gyeongbuk Institute of Science and Technology (DGIST), Daegu 42988, Korea; woori@dgist.ac.kr

**Keywords:** Ca^2+^-activated Cl^−^ channel, TMEM16A, PI(4,5)P_2_, splice variants

## Abstract

TMEM16A is a Ca^2+^-activated Cl^−^ channel that controls broad cellular processes ranging from mucus secretion to signal transduction and neuronal excitability. Recent studies have reported that membrane phospholipid phosphatidylinositol 4,5-bisphosphate (PI(4,5)P_2_) is an important cofactor that allosterically regulates TMEM16A channel activity. However, the detailed regulatory actions of PIP_2_ in splice variants of TMEM16A remain unclear. Here, we demonstrated that the attenuation of membrane phosphoinositide levels selectively inhibited the current amplitude of the TMEM16A(ac) isoform by decreasing the slow, but not instantaneous, Cl^−^ currents, which are independent of the membrane potential and specific to PI(4,5)P_2_ depletion. The attenuation of endogenous PI(4,5)P_2_ levels by the activation of *Danio rerio* voltage-sensitive phosphatase (Dr-VSP) decreased the Cl^−^ currents of TMEM16A(ac) but not the TMEM16A(a) isoform, which was abolished by the co-expression of PIP 5-kinase type-1γ (PIPKIγ). Using the rapamycin-inducible dimerization of exogenous phosphoinositide phosphatases, we further revealed that the stimulatory effects of phosphoinositide on TMEM16A(ac) channels were similar in various membrane potentials and specific to PI(4,5)P_2_, not PI4P and PI(3,4,5)P_3_. Finally, we also confirmed that PI(4,5)P_2_ resynthesis is essential for TMEM16A(ac) recovery from Dr-VSP-induced current inhibition. Our data demonstrate that membrane PI(4,5)P_2_ selectively modulates the gating of the TMEM16A(ac) channel in an agonistic manner, which leads to the upregulation of TMEM16A(ac) functions in physiological conditions.

## 1. Introduction

TMEM16A (also known as Anoctamin 1) is a Ca^2+^-activated Cl^−^ channel (CaCC) that is activated in response to intracellular Ca^2+^ increases and membrane depolarization [1,2,3]. TMEM16A is broadly expressed in tissue and organs, and plays important roles in a diverse range of physiological functions, including smooth muscle contraction, the secretion of mucus in epithelial cells, nociception, the modulation of neuronal excitability, and cell proliferation, by linking to the Ca^2+^ signaling pathway [4,5].

The TMEM16A channel is a homodimeric protein complex consisting of two identical subunits with 10 transmembrane domains (TMDs). The cytoplasmic *N*- and *C*-termini and the intracellular loops of TMEM16A possess critical regions that influence channel regulation, including dimerization [6], phosphorylation [7], and interactions with phosphoinositides such as phosphatidylinositol 4,5-bisphosphate [PI(4,5)P_2_] [8,9,10]. In particular, there are four exon segments—a (116 residues), b (22 residues), c (4 residues), and d (26 residues)—in the *N*-terminus and first intracellular loop of TMEM16A. Therefore, it is well established that several alternatively spliced TMEM16A isoforms are generated by the combinational exclusion or inclusion of each exon [11]. In addition to the four segments, the formation of other minor isoforms by Exons 1, 10, 14, and 18 has been reported [12]. Since these isoforms are differently affected by intracellular Ca^2+^ and membrane voltage, the channels in various tissues display different electrophysiological properties.

Phosphoinositides are asymmetrically distributed through the inner leaflet of the plasma membrane, where phosphatidylinositol 4-phosphate (PI4P) and phosphatidylinositol 4,5-bisphosphate [PI(4,5)P_2_] are relatively abundant compared with phosphatidylinositol 3,4,5-trisphosphate [PI(3,4,5)P_3_] [13,14,15,16]. The levels of diverse phosphoinositides in the subcellular membrane compartments are dynamically controlled by the enzymatic metabolism through lipid kinases and lipid phosphatases [17,18]. Several recent studies have revealed that PI(4,5)P_2_ is required for the open-state stabilization of TMEM16A [19,20]. For TMEM16A channels to be fully opened, the binding of PI(4,5)P_2_ and Ca^2+^ is important [21]. PI(4,5)P_2_ regulation has also been characterized as occurring through multiple PI(4,5)P_2_-binding sites in cytoplasmic loops [8,9]. In fact, the first intracellular TM2–TM3 loop contains several important regulatory domains such as the c-segment and PI(4,5)P_2_-binding region. In our recent study, we found that those PI(4,5)P_2_ binding sites in TMEM16A are allosterically modulated by the distant phosphorylation of S673 by Ca^2+^/calmodulin-dependent protein kinase II (CaMKII) [10].

The relationship between phosphoinositide dynamics and TMEM16A channel activity has been tested through experiments perfusing water-soluble exogenous dioctanoylglycerol-phosphoinositides (diC8-PIs) into the intracellular side [9,19]. Tempo et al. [19] reported that diC8-PIs strongly recovered TMEM16A current decay induced by high Ca^2+^ in the order of diC8-PI(4,5)P_2_ > diC8-PI4P >> diC8-PI. Yu et al. [9] compared the effects of diC8-poly-PIs on TMEM16A currents and found diC8-PI(3,4,5)P_3_ to be less efficacious than diC8-PI(4,5)P_2_ at the same concentrations, and diC8-phosphatidylinositol 3,5-bisphosphate [diC8-PI(3,5)P_2_]) was found to have no significant stimulatory effect. Studies on this topic have commonly observed that the addition of diC8-PI(4,5)P_2_ rescues the TMEM16A-induced Cl^−^ current from rundown and increases the TMEM16A current more strongly compared with other diC8-PIs. However, the effects of diverse phosphoinositides on TMEM16A activity under endogenous cell systems have not been clearly determined. Here, using various exogenous lipid phosphatase tools that can directly cleave the indicated phosphate group from phosphoinositides without generating any other secondary messengers, we verified that the TMEM16A channel is selectively inhibited by the depletion of endogenous PI(4,5)P_2_ but no other phosphoinositides in live cells. We also found that PI(4,5)P_2_ sensitivity differed between fast and slow TMEM16A currents, which were maintained similarly throughout various membrane voltages. Together, our data demonstrated that membrane PI(4,5)P_2_ differentially but finely tunes the biophysical properties of alternatively spliced TMEM16A channel gating.

## 2. Results

### 2.1. Different Sensitivities of TMEM16A(a) and TMEM16A(ac) to PI(4,5)P_2_ Depletion

Previous studies have reported that TMEM16A splice variants possessing c-segments showed relatively higher sensitivity to intracellular Ca^2+^ concentrations [22,23]. Here, we used two TMEM16A isoforms that had an a-segment only or both the a- and c-segments (Figure 1A). Using the zebrafish voltage-sensing phosphatase (Dr-VSP), we first investigated how the splice variant TMEM16A(ac) responded to PI(4,5)P_2_ regulation. Lipid phosphatase was activated by the depolarization of the membrane potential and quickly removed the 5-phosphate from PI(4,5)P_2_ [24,25]. In the control experiments without Dr-VSP, there was no significant difference in the peak plateau current amplitudes (blue arrowhead) before (Trace a) and after (Trace b) a 1 s depolarization to +120 mV for both the TMEM16A(a) and TMEM16A(ac) currents. The strong depolarizing pulse attenuated the subsequent TMEM16A(ac) current by 23 ± 1% in cells expressing Dr-VSP (Figure 1B,C). However, it did not significantly attenuate the TMEM16A(a) current. To test whether the inhibition of TMEM16A(ac) by Dr-VSP was caused by PI(4,5)P_2_ depletion, the effect of Dr-VSP activation was measured in cells co-expressing PIP 5-kinase type-1γ (PIPKIγ), which accelerates synthesis and elevates the PI(4,5)P_2_ concentration in the cell membrane [26]. Co-expression with PIPKIγ almost completely abolished the current inhibition induced by 1-s Dr-VSP activation (Figure 1B,D).

### 2.2. Different PI(4,5)P_2_ Sensitivities in Instantaneous and Slow TMEM16A(ac) Currents

Under moderate intracellular Ca^2+^ levels, TMEM16A activation traces can be divided into two conductance phases: a small, rapid, instantaneous current (*I*_Inst_) and a slowly activating current (*I*_Slow_) with a subsequent steady-state level [23,27,28]. To identify whether both the instantaneous and slow components of the TMEM16A current are manipulated by PI(4,5)P_2_ regulation, we analyzed the PI(4,5)P_2_ depletion-mediated current suppression in the two gating phases in the presence of 115 nM cytosolic Ca^2+^ (Figure 2A). We found that the current reductions in TMEM16A(ac) by PI(4,5)P_2_ depletion were 5 ± 1% and 34 ± 1% in instantaneous and slow currents, respectively (Figure 2B), indicating that the two phases have different sensitivities to PI(4,5)P_2_ and that PI(4,5)P_2_ regulation is much more prominent in the slower steady-state currents (*I*_Slow_) of TMEM16A(ac). No significant differences were observed for TMEM16A(a) between the two components (Figure 2B).

### 2.3. Membrane Potential-Independent Inhibition of TMEM16A Channels by PI(4,5)P_2_

To further investigate the effects of PI(4,5)P_2_ on TMEM16A currents in living cells, we applied the rapamycin-inducible translocation system. In this system, rapamycin forms a ternary complex with FKBP (FK506 binding protein) and FRB (the rapamycin-binding domain of mTOR). Here, we used the fusion proteins of mRFP-FKBP (RF) with two lipid phosphatase enzymes [29]; 4-phosphatase (Sac1) dephosphorylates PI4P at Position 4 of the inositol ring and inositol pholyphosphate-5-phosphatase E (INPP5E) dephosphorylates PI(4,5)P_2_ at Position 5 of the inositol ring. By combining either the inactive or active forms of each enzyme, there were 2 constructs, as in Figure 3A: RF-Dead, with no activity, and pseudojanin (PJ), with both Sac1 and INPP5E phosphatase. Upon the application of rapamycin, these proteins can be recruited to a plasma membrane anchor Lyn11-FRB comprising LDR and the 11 *N*-terminal sequences of Lyn kinase (including its myristylation and palmitoylation sites and basic amino acids) coupled to an FRB domain [29]. To monitor the specific enzymatic activity of the translocatable phosphatase, we used a genetically expressed fluorescent PI(4,5)P_2_ probe, the pleckstrin homology (PH) domain from PLCδ1 (CFP-PH-PLCδ1). The CFP-PH-PLCδ1 probe marks PI(4,5)P_2_ at the plasma membrane in resting cells and migrates into the cytosol when PI(4,5)P_2_ is depleted. The translocation of RF-PJ (but not RF-Dead) by rapamycin selectively initiated the migration of the PI(4,5)P_2_ probe, which is visible as an increase in the fluorescence intensity in the cytosol (blue trace in Figure 3B,C). These results are consistent with the consequences published previously [30].

Using these two tools, we examined the changes in the TMEM16A current after treatment with 1 μM rapamycin for 60 s. Recruiting RF-Dead had no effect on either TMEM16A(ac) current. In contrast, recruiting RF-PJ significantly inhibited the currents by 18 ± 2%, which suggests that PI(4,5)P_2_ is required for the activation of TMEM16A (Figure 3D,E). However, with the TMEM16A(a) splice variant, the translocation of neither RF-Dead nor RF-PJ affected the current (Figure 3D,E). To examine whether the current inhibition by PI(4,5)P_2_ depletion was affected by the applied voltages, we measured the effects of Dr-VSP activation on TMEM16A currents at positive potentials between +20 and +50 mV, which triggered the outward rectification of currents. The results showed no significant voltage dependence for the effects of rapamycin-induced translocation on the TMEM16A(a) and TMEM16A(ac) channels (Figure 3F), demonstrating that the effects of rapamycin-induced translocation on the TMEM16A channels are independent of the membrane potential.

### 2.4. Phosphoinositide Sensitivity of TMEM16A Is Specific for PI(4,5)P_2_

In addition to PI(4,5)P_2_, other phosphoinositides such as PI4P or PI(3,4,5)P_3_ have been reported to modulate the activity of ion channels [31,32,33,34]. To validate phosphoinositide specificity in TMEM16A, we further performed experiments with diverse rapamycin-induced dimerization systems labeled with either active Sac1, INPP5E, or PTEN (a phosphatase and tensin homolog) that selectively de-phosphatize PI4P, PI(4,5)P_2_, and PI(3,4,5)P_3_, respectively (Figure 4A). We found rapamycin to have no effect on the TMEM16A current in cells expressing RF-Sac1 (Figure 4B, top). However, rapamycin-induced recruitment of RF-INPP5E decreased the TMEM16A current by 14 ± 1% (Figure 4B,C, top), which suggests that PI4P is unable to regulate TMEM16A activity. For PI(3,4,5)P_3_, both the control CFP-FKBP (CF) and CFP-FKBP-PTEN (CF-PTEN) showed no changes in TMEM16A current after rapamycin application (Figure 4B,C, bottom). The results regarding the PI(3,4,5)P_3_ effects are consistent with the observations by the Tian group, who found that siRNA knockdown of PTEN or PI3-kinase inhibition by wortmannin did not affect TMEM16A activation by ionomycin [35]. Collectively, our results provide evidence that among the membrane phosphoinositides, PI(4,5)P_2_ specifically affects the activation of TMEM16A in living cells.

### 2.5. PI(4,5)P_2_ Resynthesis Is Required for Current Recovery from Dr-VSP-Induced Inhibition

Next, we tested whether recovery from Dr-VSP-induced inhibition requires PI(4,5)P_2_ resynthesis. The rate of recovery was measured with a voltage protocol reported previously [36]. The control channel current was measured by a +50-mV test pulse and the recovery after the activation of Dr-VSP was as shown in the protocol (Figure 5A). The TMEM16A(ac) current recovered to the initial level with a time constant (τ) of 4.6 s. However, recovery was almost completely blocked in cells perfused with the nonhydrolyzable ATP analog AMP-PNP (Figure 5B). In the cells co-expressing PIPKIγ, the current inhibition of TMEM16A(ac) was dramatically reduced. The prevention of recovery by AMP-PNP and the prevention of inhibition by overexpressed PIPKIγ confirmed that TMEM16A(ac) needs PI(4,5)P_2_ for maximal activation and PI(4,5)P_2_ resynthesis for recovery from Dr-VSP-induced inhibition.

### 2.6. TMEM16A(ac) Current Modulation by M_1_ Muscarinic Receptor Activation

When the G_q_ protein-coupled receptors (G_q_-GPCRs) are activated, membrane PI(4,5)P_2_ is hydrolyzed into diacylglcerol and inositol-1,4,5-trisphosphate by phospholipase C. Thus, previous studies reported that PI(4,5)P_2_-sensitive ion channels are modulated upon the activation of G_q_-GPCRs [36,37]. To test this possibility in TMEM16A(ac) channels, we measured the effects of M_1_ muscarinic receptor (M_1_R) activation on the current amplitude in cells expressing TMEM16A and M_1_R. To exclude the effect of intracellular Ca^2+^ rise produced by M_1_R activation, cells were dialyzed with a pipette solution containing zero Ca^2+^ plus the Ca^2+^ chelator EGTA (See the Methods section for detailed information). TMEM16A can be activated by depolarizing pulses of more than +100 mV in the absence of intracellular Ca^2+^ [22]. Based on this point, we performed a whole-cell recording of TMEM16A(ac) at +100 mV in cells intracellularly dialyzed with a pipette solution containing EGTA and zero Ca^2+^. As shown in Figure 6, the activation of M_1_R with Oxo-M significantly inhibited TMEM16A(ac) currents, while no significant reduction in currents was observed in cells expressing only TMEM16A(ac) without M_1_R, suggesting that G_q_-GPCR activation negatively modulates TMEM16A(ac) gating through the PI(4,5)P_2_ cleavage. Since M_1_R activation elevates intracellular Ca^2+^ concentrations through the hydrolysis of membrane PI(4,5)P_2_, G_q_-GPCR-mediated TMEM16A(ac) channel regulation is likely to be the result of physiological tuning processes between Ca^2+^-induced activation and PI(4,5)P_2_ cleavage-induced inhibition in living cells.

## 3. Discussion

Various isoforms of TMEM16A channels can be generated through the combination of four alternatively spliced exon segments, a, b, c, and d. Previous studies reported that the inclusion or exclusion of b- and c-segments particularly affected the channels’ Ca^2+^ and voltage sensitivity [11]. Our recent study showed that the inclusion of the c-segment potentiated the current density and PI(4,5)P_2_ sensitivity of TMEM16A channels. In this study with HEK293T cells, we found some new features of PI(4,5)P_2_ regulation in TMEM16A channels: (1) by using diverse exogeneous lipid phosphatase systems, we confirmed that the TMEM16A channel is regulated solely by the levels of PI(4,5)P_2_ and no other phosphoinositides in the plasma membrane; (2) of the two separate phases of TMEM16A activation, the slow current (*I*_Slow_) phase is mainly affected by PI(4,5)P_2_ depletion; (3) PI(4,5)P_2_ depletion-induced current inhibition is independent of the membrane potential; (4) PI(4,5)P_2_ resynthesis using cytosolic ATP is required for the recovery of the TMEM16A current from Dr-VSP-mediated suppression.

Our data showed how membrane PI(4,5)P_2_ turnover modulates the TMEM16A channels in living cell membranes and revealed novel differences between the two isoforms of TMEM16A channels in the modulation by PI(4,5)P_2_. Our recent study revealed that the c-segment allosterically regulates the interaction of TMEM16A channels with PI(4,5)P_2_ by altering the structure of PI(4,5)P_2_-binding sites on the channel protein, while it does not bind to PI(4,5)P_2_ directly [10]. The activity of the TMEM16A(ac) channels is selectively decreased by the conversion of PI(4,5)P_2_ to PI4P and remains inhibited until PI(4,5)P_2_ is resynthesized. Current recovery from Dr-VSP-induced inhibition occurs through the metabolic resynthesis of PI(4,5)P_2_ from PI4P by a phosphorylation reaction using intracellular ATP. According to our results, the time constant for TMEM16A current recovery after Dr-VSP activation is 4.6 s. Since Dr-VSP activation simply converts PI(4,5)P_2_ to PI4P, the time constant for current recovery indicates the time for the resynthesis of PI(4,5)P_2_ from PI4P by endogenous PIP 5-kinases. Our data indicate that PI(4,5)P_2_ resynthesis from PI4P requires intracellular hydrolyzable ATP and occurs much faster than the resynthesis of PI(4,5)P_2_ from PI [38]. Ta et al. reported the time constant for recovery to be 7.2 s [39]. This minor disagreement in recovery time may have been caused by differences in the duration of the depolarizing pulses applied. They conducted the recovery experiment using a double-pulse protocol, where two +100 mV/4 s depolarizing pulses were given at different time intervals. In our study, after applying a single +120 mV/1 s depolarizing pulse, the recovery of TMEM16A from inhibition was measured by applying various interval times. Previous studies showed that the activation of Dr-VSP for less than 1 s with +100–120 mV depolarizing pulses is enough for depleting membrane PI(4,5)P_2_ [36]. Therefore, because of the rapid turnover between the phosphoinositides in cell membranes, the increase in the depolarizing duration would deplete other poly-phosphoinositides, such as PI4P and PI(3,4,5)P_3_, which might retard the resynthesis of PI(4,5)P_2_ and current recovery from the inhibition.

The outcomes of PIPKIγ overexpression shown in Figure 1 and current recovery in Figure 5 are consistent with the consequences demonstrated by the Ta group [39]. PIPKIγ overexpression in cells increases the amount of PI(4,5)P_2_ in the membrane; under these conditions, Dr-VSP activation does not significantly affect the changes in the TMEM16A current, unlike the control condition. Overall, the anionic phosphoinositide PI(4,5)P_2_ is a required cofactor for the full channel activity of TMEM16A(ac). Since the channels must be in equilibrium with the plasma membrane pools of PI(4,5)P_2_ on a much shorter time scale than the 100 ms it takes for Dr-VSP to depress their currents [36], the binding affinity between the channel protein and the PI(4,5)P_2_ lipid must be very low.

A single TMEM16A current displays a biphasic activation trace with instantaneous and slow-activating components because of the difference in its gating properties in response to Ca^2+^-binding and voltage changes [28]. A TMEM16A channel possesses two Ca^2+^-binding sites with different binding affinities. In the presence of moderate intracellular Ca^2+^ concentrations, a small number of channels are bound to a single Ca^2+^. After depolarization, the few channels with a single Ca^2+^ occupancy open quickly and allow Cl^−^ flux, which then presents the small, rapid *I*_Inst_. As time passes, the number of TMEM16A channels combined with single Ca^2+^ increases and, upon membrane depolarization, these channels favor the second Ca^2+^ coupling, which leads to the activation of *I_Slow_* slowly [28]. In our experiments with a Ca^2+^ concentration of 115 nM, *I*_Inst_ was prominently measured before *I*_Slow_. However, we found that *I*_Inst_ was much less sensitive to PI(4,5)P_2_ depletion than the slow *I*_Slow_. This may be due to the conformation of differential channel structures by two sequential bindings of Ca^2+^. Membrane PI(4,5)P_2_ would be more important for maintaining the channels in the fully open state with two Ca^2+^-binding sequences. Similarly, we report that the modulatory effects of PI(4,5)P_2_ are stronger in unphosphorylated TMEM16A(ac) channels [10], suggesting that the binding of PI(4,5)P_2_ to TMEM16A channels is dynamically regulated by allosteric changes in the channel structure.

In this study, we used the rapamycin-induced dimerization system in addition to Dr-VSP among the tools that can confirm the phosphoinositide’s effects without a second messenger. Rapamycin-induced dimerization provides us two new pieces of information: (1) the inhibitory effects of the TMEM16A(ac) current by PI(4,5)P_2_ depletion are independent of membrane potential; and (2) the dephosphorylation of 5′-phosphate—but not 3′- or 4′-phosphate—in the phosphatidylinositol ring affects the specific inhibition of the TMEM16A Cl^−^ current. Consistent with our findings, recent reports have shown that the application of diC8-PI(4,5)P_2_—a soluble PI(4,5)P_2_ analog—potentiates the Cl^−^ current in TMEM16A and recovers from current rundown [9,19]. However, Tembo et al. [19] and Yu et al. [9] showed that TMEM16A channel activity is slightly enhanced by the application of 100 μM diC8-PI4P and 10 μM diC8-PI(3,4,5)P_3_, respectively. The discrepancy between those results and our data is likely to have been caused by the accumulation of exogenous phosphoinositides to supramaximal levels in the plasma membrane. Similarly, in comparative experiments performed with receptor stimulation, voltage-sensitive lipid 5-phosphatase, or engineered fusion protein carrying phosphatase, Kruse et al. [40] found that the regulation of K_V_7 channels by these three methods was inconsistent with other groups who applied exogenous PI(4,5)P_2_. They speculated that one of the possibilities is due to superphysiological high concentrations of phosphoinositide in cells when exogenous phosphoinositide was applied.

According to our results, the inhibition ratio of TMEM16A(ac) currents by the translocation of PJ was slightly greater than that of INPP5E (Figure 3E and Figure 4C). This might be because of the rapid turnover between PI4P and PI(4,5)P_2_ [41,42]. PI(4,5)P_2_ is continuously and rapidly reproduced from PI4P by PI4P 5-kinase in the plasma membrane [42,43]. PJ containing both 4′-phosphatase Sac1 and 5′-phosphatase INPP5E can dephosphorylate both PI(4,5)P_2_ and PI4P simultaneously, whereas INPP5E with only 5′-phosphatase does not deplete PI4P. Since PI4P is the precursor of PI(4,5)P_2_, PI(4,5)P_2_ resynthesis in the plasma membrane can be faster in experiments with INPP5E. This may be the cause of why the inhibition of TMEM16A(ac) currents by INPP5E is lower compared with PJ.

Of the diverse phosphoinositides detected in the inner leaflet of the plasma membrane, PI(4,5)P_2_ is usually the main poly-phosphoinositide [44]. PI(4,5)P_2_ constitutes 0.2–1% of the total cellular membrane lipids and 2–5 mol% of the total phosphoinositides in the plasma membrane [43,45]. According to this information, PI(4,5)P_2_ is the second largest phosphoinositide behind PI in the plasma membrane of most mammalian cells. However, the composition of these phosphoinositides is likely to vary depending on cell types. It was reported that PI(4,5)P_2_ levels are 5–10 times higher in intact cardiac tissue than in isolated myocytes [46]. For this reason, the level of PI(4,5)P_2_ does not change significantly during the activation of muscarinic receptors in intact atrial tissue [47]. Therefore, the effects of PI(4,5)P_2_ on TMEM16A may also be different depending on the phosphoinositide composition of the cells.

The alternative splicing patterns of TMEM16A are different for each human tissue. The exon b-segment shows the highest expression rate in the liver and thyroid, and the expression rate is >70% in the placenta, prostate, and trachea. The d-segment is mainly present in adipose, brain, cervix, colon, heart, kidney, lung, ovary, small intestine, and thymus tissues. Although the level of mRNA is somewhat lacking in the brain and skeletal muscles, the c-segment shows a higher percentage of inclusion than other segments in diverse human tissues, including the heart, kidney, and liver [11]. Given that TMEM16A’s alternative splicing is expressed in a tissue-specific manner, it is important to comprehend the PI(4,5)P_2_ regulation of TMEM16A.

## 4. Materials and Methods

### 4.1. Cell Culture and Transfection

HEK293T cells (large T-antigen-transformed HEK293 cells) were maintained in Dulbecco’s modified Eagle’s medium (DMEM) (HyClone, Thermo Fisher Scientific, Waltham, MA, USA) supplemented with 10% fetal bovine serum (HyClone, Thermo Fisher Scientific) and 0.2% penicillin/streptomycin (HyClone, Thermo Fisher Scientific) in 100 mm culture dishes at 37 °C with 5% CO_2_. For TMEM16A expression, 500 ng of GFP-TMEM16A was transiently transfected into HEK293T cells using Lipofectamine 2000 (Invitrogen, Carlsbad, CA, USA) per 35 mm plate at 50–60% confluency in all experiments. In the PI(4,5)P_2_ sensitivity experiments, cells were co-transfected with 1000 ng Dr-VSP and 800 ng PIPKIγ. For the rapamycin-inducible dimerization experiment, cells were co-transfected with 300 ng of Lyn11-FRB, translocatable enzymes (RF-Dead and RF-PJ), and CFP-PH-PLCδ1. Transfected cells were plated onto poly-L-lysine (0.1 mg/mL, Sigma-Aldrich, St. Louis, MO, USA)-coated chips and used for voltage-clamp recordings and imaging experiments at 24–36 h after transfection.

### 4.2. Plasmid and Chemical

Mouse cDNA clones of TMEM16A(ac) (GenBank Accession No. AAH_62959.1) were generously given by Frank H. Yu (University of Seoul, Seoul, Korea). The Dr-VSP was given by Yasushi Okamura (Osaka University, Osaka, Japan). The rapamycin-inducible dimerization system, RF-Dead, RF-Sac1, RF-INPP5E, RF-PJ, and Lyn11-FRB (LDR) were provided by Takanari Inoue (Johns Hopkins University, Baltimore, MD, USA) and Gerald R. Hammond (University of Pittsburgh School of Medicine, Pittsburgh, PA, USA). The PH-RFP (PLCδ1) was from Ken Mackie (University of Washington, Washington, DC, USA). PIPKIγ was provided by Yoshikatsu Aikawa and Thomas F. Martin (University of Wisconsin, USA). The following compounds were obtained: AMP-PNP (Roche) and rapamycin (Sigma-Aldrich, St. Louis, MO, USA).

### 4.3. Solutions

For whole-cell patch configuration and confocal imaging of TMEM16A, the following extracellular solution was used (in mM): 150 NaCl, 1 CaCl_2_, 1 MgCl_2_, 10 glucose, and 10 HEPES; the pH was adjusted to pH 7.4 with NaOH. The pipette (intracellular) solution contained (in mM): 130 CsCl, 1 MgCl_2_, 10 EGTA, and either 3 Na_2_ATP or 3 AMP-PNP. For the standard pipette solution, 5.83 and 8.47 mM CaCl_2_ were added to make a free [Ca^2+^]_i_ of 115 and 455 nΜ (calculated with the Ca/Mg/ATP/EGTA calculator v2.2b available at https://somapp.ucdmc.ucdavis.edu/pharmacology/bers/maxchelator/webmaxc/webmaxcS.htm, accessed on 3 July 2009) adjusted to pH 7.35 with CsOH. To test TMEM16A modulation by the M_1_ muscarinic receptor, CaCl_2_ was omitted from the intracellular solution but 0.1 mM Na_3_GTP was supplemented instead.

### 4.4. Current Recording

Whole-cell patch-clamp recordings were performed as described previously [10]. We used a HEKA EPC-10 amplifier with Pulse software (HEKA Elektronik) for the acquisition of Cl^−^ currents. Patch pipettes with resistances of 2–5 MΩ were pulled from borosilicate glass micropipette capillaries (Sutter Instrument Co., Novato, CA, USA) using a Flaming/Brown micropipette puller (P-97, Sutter Instrument Co.). Series resistance errors were compensated by >60%, and the fast and slow capacitances were compensated before the application of test pulses. TMEM16A currents were recorded with a membrane-holding potential of −60 mV and the application of a 500 ms test pulse. During recording, we used a 6-channel valve controller system (VC-6, Warner Instruments, Holliston, MA, USA) to deliver the external solution to the cells placed on a Quick Change Chamber Narrow Slotted Bath (RC-46SNLP, Warner Instruments). The complete solution exchange was achieved within one second. For Dr-VSP experiments, the protocol used was as follows. TMEM16A was activated by test Pulse a (+50 mV) for 500 ms. Next, step depolarizations to 120 mV were applied for 1 s to activate Dr-VSP and deplete the PI(4,5)P_2_ in cells. After the application of a high depolarizing pulse, a –150 mV hyperpolarizing pulse was applied followed by test Pulse b (+50 mV) for 500 ms. For data acquisition and analysis, we used Pulse/Pulse Fit software combined with an EPC-10 patch clamp amplifier (HEKA Elektronik, Pfalz, Germany) and Igor Pro (WaveMetrics, Inc., Tigard, OR, USA).

### 4.5. Confocal Imaging and Quantitation

HEK293T cells were imaged 1–2 days after transfection on poly-L-lysine-coated chips with a Carl Zeiss LSM 700 or LSM 800 confocal microscope (Carl Zeiss AG, Jena, Germany) at room temperature. Cell images were scanned using a 40× (water) objective lens at 1024 × 1024 pixels using a digital zoom. For the time course experiments, 512 × 512 pixels were used. Image processing was carried out using Zeiss ZEN 2.3 SP1 software. Cytosolic fluorescence intensity in the time course experiments was assessed by forming relative values from the regions of interest drawn in the cytoplasmic region. PH-PLCδ1 and translocatable enzymes were normalized to the minimum and maximum intensities, respectively. All images were transferred from the LSM4 to JPEG format. The raw data were processed in Excel 2016 (Microsoft) and Igor Pro (WaveMetrics, Inc., Tigard, OR, USA).

### 4.6. Statistical Analysis

All data were analyzed using Excel 2016 (Microsoft Inc., Redmond, WA, USA), IGOR Pro 6.0 (WaveMetrics, Inc., Tigard, OR, USA), or GraphPad Prism 7.02 (GraphPad Software Inc., San Diego, CA, USA). Statistics in text or figures represent means ± SEM. Statistical comparisons between the two groups were analyzed using Student’s *t*-tests. The significance of the observations among more than two groups was assessed by one-way ANOVA followed by Sidak’s post hoc test. Differences were considered significant at the * *p* < 0.05, ** *p* < 0.01, and *** *p* < 0.001 levels.

## 5. Conclusions

This study expands our understanding of PI(4,5)P_2_ regulation of the TMEM16A channels in living cells. Using electrophysiological recordings and confocal microscopy, the present works show that two splice variants, TMEM16A TMEM16A(a) and TMEM16A(ac), are differentially regulated by the membrane lipid PI(4,5)P_2_. Our results suggest that the slow component of TMEM16A(ac) currents is selectively inhibited by the attenuation of plasma membrane PI(4,5)P_2_ levels, and co-expression of PIPKIγ eliminates the inhibition by elevating PI(4,5)P_2_ synthesis. In the chemical-induced dimerization assays using exogenous phosphoinositide phosphatases, we confirmed that TMEM16A(ac) current reduction appears only with the dephosphorylation of PI(4,5)P_2_, but not other phosphoinositides, in a membrane-potential-independent manner. Additionally, we observed that the PI(4,5)P_2_ depletion-mediated current inhibition occurs during G_q_-coupled receptor activation. These observations corroborate the role of PI(4,5)P_2_ as an important modulatory factor of TMEM16A channel gating.

## Figures and Tables

**Figure 1 ijms-22-04088-f001:**
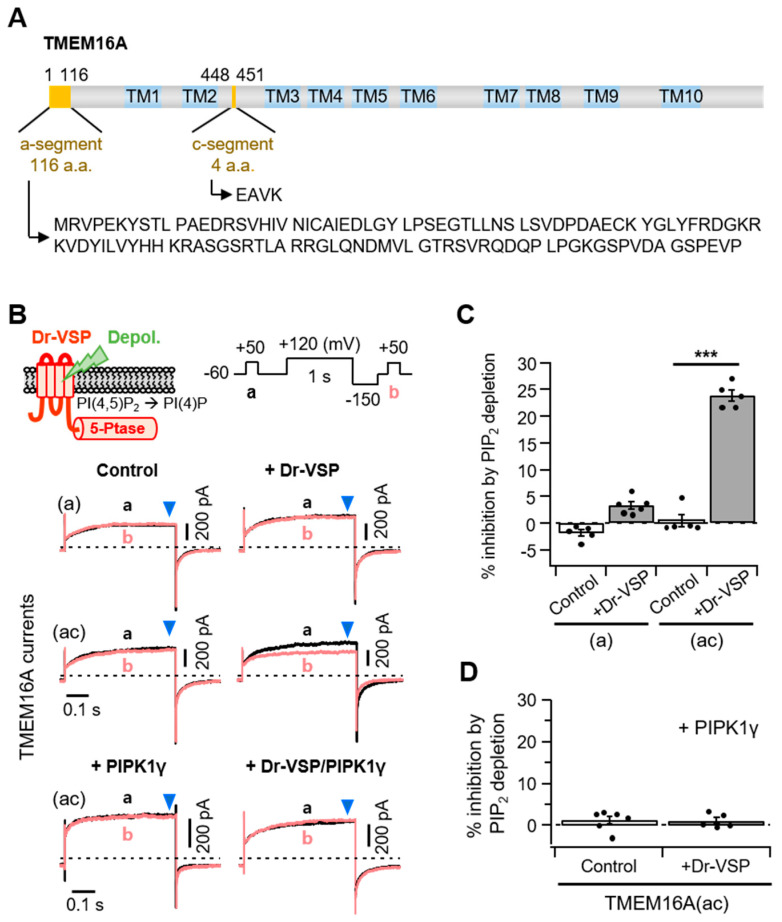
Different PI(4,5)P_2_ sensitivities of alternatively spliced TMEM16A(a) and TMEM16A(ac) channels. (**A**) Domain-architecture schematics of the TMEM16A(ac) channel consisting of two splicing segments (yellow box) and 10 transmembrane domains (TM; blue box). The numbers above the diagram indicate the amino acid positions of two segments, “a” and “c”, in TMEM16A. The amino acid sequences for the alternatively spliced “a” (116 residues) and “c” (four residues) are shown below. (**B**) Top: An illustration of Dr-VSP-mediated PI(4,5)P_2_ depletion in the plasma membrane. The inset shows the voltage protocol with a large depolarization for activating the Dr-VSP. Bottom: A comparison between the inhibition of TMEM16A(a) and TMEM16A(ac) currents by membrane depolarization in the control and Dr-VSP-expressing cells. The currents at +50 mV before (a, black) and after (b, red) 1 s depolarizing pulses to +120 mV are superimposed. The pipette solution contained 3 mM ATP and 455 nM (TMEM16A(a)) or 115 nM (TMEM16A(ac)) [Ca^2+^]_i_. (**C**) Summary of TMEM16A(a) and TMEM16A(ac) current inhibition (%) by membrane depolarization in the control and Dr-VSP-expressing cells. *n* = 5. *** *p* < 0.001, one-way analysis of variance (ANOVA) followed by Sidak’s post hoc test. (**D**) Summary of current inhibition (%) of TMEM16A(ac) by membrane depolarization in cells expressing PIPKIγ alone or Dr-VSP plus PIPKIγ. *n* = 5–6. Dots indicate the individual data points for each cell. Bars indicate means ± standard error of the mean (SEM).

**Figure 2 ijms-22-04088-f002:**
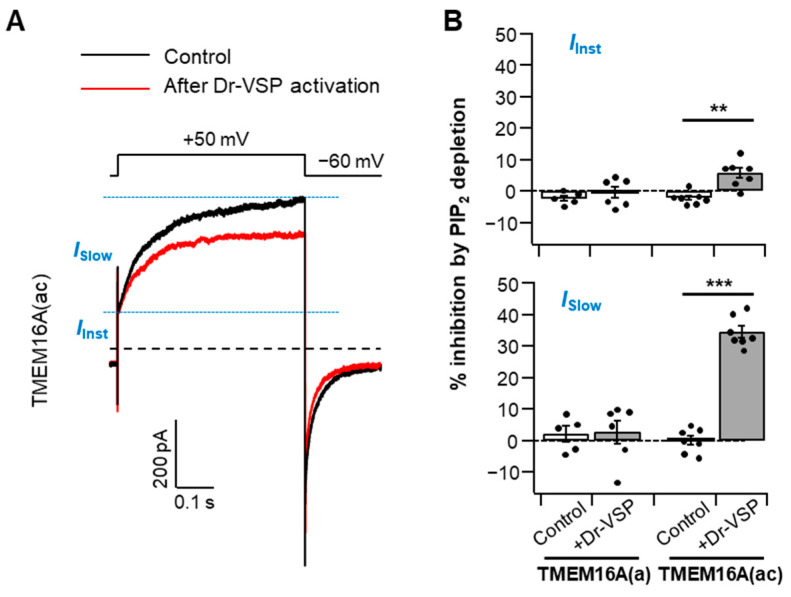
Analysis of PI(4,5)P_2_ regulation regarding the two steps of TMEM16A activation. (**A**) Representative Cl^−^ current traces activated by a voltage step to +50 mV before (control) and after Dr-VSP activation in cells expressing TMEM16A(ac) in the presence of a 115 nM intracellular Ca^2+^ concentration. Black trace: control (before Dr-VSP activation). Red trace: after Dr-VSP activation. The dotted line indicates zero current. The horizontal dashed lines (blue) indicate the separation point for the instantaneous and slow currents. *I_Inst_*—instantaneous Cl^−^ current; *I_Slow_*—slow Cl^−^ current. (**B**) Inhibition (%) of the instantaneous (top) and slow (bottom) currents in response to PI(4,5)P_2_ depletion by Dr-VSP activation in cells expressing TMEM16A(a) or TMEM16A(ac); *n* = 5–7. Dots indicate the individual data points for each cell. Bars indicate means ± SEM. ** *p* < 0.01, *** *p* < 0.001, one-way ANOVA followed by Sidak’s post hoc test.

**Figure 3 ijms-22-04088-f003:**
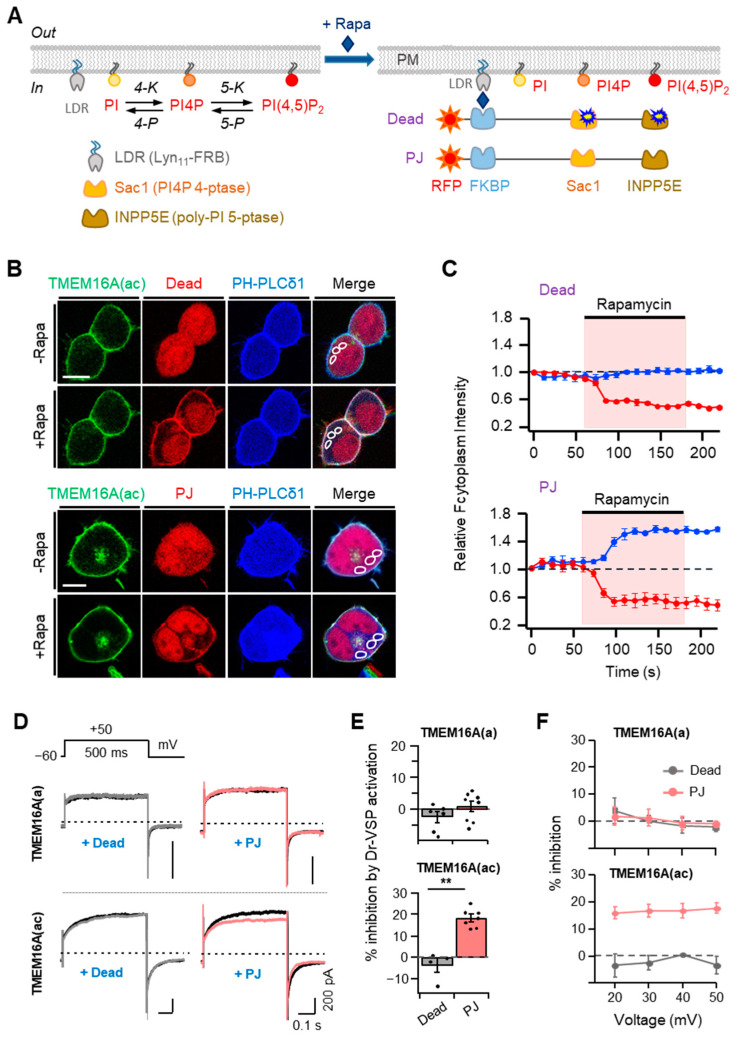
Effects of rapamycin-induced translocation of poly-phosphoinositide phosphatases on TMEM16A currents. (**A**) Left: metabolic pathway of PI(4,5)P_2_ synthesis by lipid kinases (4-K and 5-K) and breakdown by lipid phosphatases (4-P and 5-P). PI, phosphatidylinositol; PI4P, phosphatidylinositol 4-phosphate; LDR, Lyn11-FRB; Sac1, PI4P 4-phosphatase; INPP5E, poly-PI 5-phosphatase. Right: schematic diagram showing the rapamycin-induced dimerization of FRB and FKBP proteins. This dimerization leads to the recruitment of poly-PI-metabolizing enzymes to the plasma membrane. RF-Dead is a translocatable construct with inactive mutant Sac1 and INPP5E enzymes. (**B**) Confocal images of cells expressing RF-Dead or RF-PJ with GFP-TMEM16A(ac), CFP-PH-PLCδ1, and LDR. Images were acquired before (−) and after (+) the application of rapamycin (1 μM) for 120 s. Images are representative of three to five cells in three independent experiments. In each cell, three regions of interest were marked in confocal images for the analysis of cytosolic fluorescence intensity in a single cell. The scale bar represents 10 μm. (**C**) The time course of rapamycin effects on the relative cytosolic fluorescence intensities of CFP-PH-PLCδ1 (blue) and phosphatase enzymes (red). (**D**) Representative TMEM16A(a) (top) and TMEM16A(ac) (bottom) currents before (black trace) and after (colored trace) the addition of rapamycin (1 μM) for 1 min in cells co-transfected with LDR and RF-Dead (gray) or RF-PJ (pink). (**E**) Summary of inhibition (%) in TMEM16A(a) and TMEM16A(ac) by rapamycin-induced translocation of RF-Dead or RF-PJ to the plasma membrane (TMEM16A(a): RF-Dead, *n* = 6, RF-PJ, *n* = 8; TMEM16A(ac): RF-Dead, *n* = 4, RF-PJ, *n* = 6). Dots indicate the individual data points for each cell. Bars indicate means ± SEM. ** *p* < 0.01 compared with RF-Dead. (**F**) Voltage independence of the rapamycin-induced inhibition of TMEM16A(a) and TMEM16A(ac) channels. The percent inhibition is plotted as a function of the membrane potential (mV).

**Figure 4 ijms-22-04088-f004:**
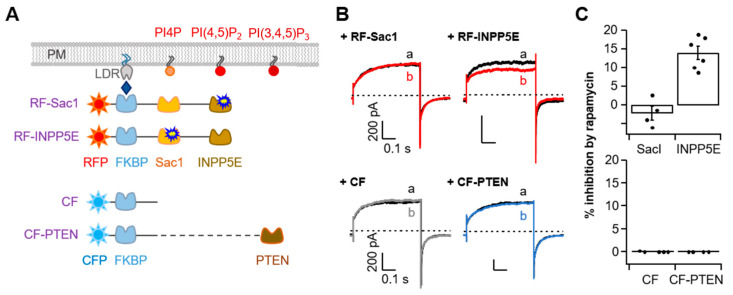
Specific TMEM16A current inhibition via the depletion of PI(4,5)P_2_ but not PI4P or PI(3,4,5)P_3_. (**A**) Strategy used to dephosphorylate PI4P, PI(4,5)P_2_, or PI(3,4,5)P_3_ using a rapamycin-induced dimerization system. RF-Sac1, 4-phosphatase; RF-INPP5E, 5-phosphatase; CF-PTEN, PI(3,4,5)P_3_ phosphatase. CF was constructed without a phosphatase enzyme. (**B**) Representative currents before (black trace) and after (colored trace) the addition of rapamycin (1 μM) for 1 min in control cells co-expressing RF-Sac1, RF-INPP5E, CF, or CF-PTEN. (**C**) Summary of inhibition (%) in (**B**) by rapamycin-induced translocation of RF-Sac1, RF-INPP5E, CF, or CF-PTEN to the plasma membrane for TMEM16A(ac). Top: RF-Sac1, *n* = 4; RF-INPP5E, *n* = 7. Bottom: CF, *n* = 4; CF-PTEN, *n* = 4. Dots indicate the individual data points for each cell. Bars indicate means ± SEM.

**Figure 5 ijms-22-04088-f005:**
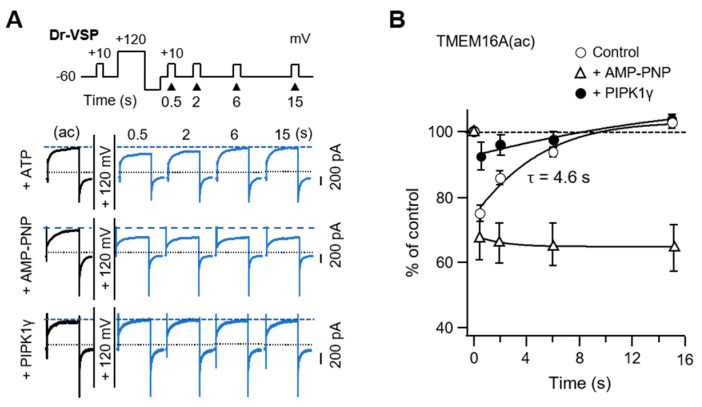
TMEM16A(ac) current recovery from inhibition by PI(4,5)P_2_ depletion. (**A**) Current traces for TMEM16A(ac) in control cells intracellularly perfused with 3 mM ATP (top), cells perfused with AMP-PNP (middle), and cells expressing PIPKIγ with 3 mM ATP (bottom) before and after a +120 mV 1 s depolarizing pulse. TMEM16A(ac) currents were measured at +50 mV at the indicated times after the depolarizing pulse. Dotted lines indicate a zero current; dashed lines indicate the initial TMEM16A(ac) current before the depolarizing step. (**B**) Summary time courses of TMEM16A(ac) current recovery in response to PI(4,5)P_2_ depletion by Dr-VSP activation (ATP, *n* = 6; AMP-PNP, *n* = 5; PIPKIγ, *n* = 5).

**Figure 6 ijms-22-04088-f006:**
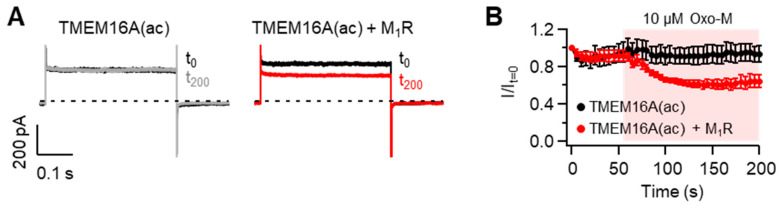
Modulation of TMEM16A(ac) channels by the M_1_ muscarinic receptor (M_1_R). (**A**) Representative Cl^−^ current traces of TMEM16A(ac) before and after application of the M_1_R agonist oxotremorine-M (Oxo-M). Whole-cell recordings were measured before (*t*_0_, black) and 200 s after (*t*_200_, gray and red) the application of 10 μM Oxo-M in cells co-transfected without (left) and with (right) M_1_R. Cells were dialyzed with an intracellular solution including 10 mM EGTA and zero Ca^2+^, and the current was measured at +100 mV every 5 s. (**B**) Normalized mean current ± SEM. TMEM16A(ac), *n* = 5; TMEM16A(ac) + M_1_R, *n* = 5.

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
