# Peer review of "Differential Regulation of Ca^2+^-Activated Cl^−^ Channel TMEM16A Splice Variants by Membrane PI(4,5)P_2"

_ijms, 2021, doi:10.3390/ijms22084088_

Round 1

Reviewer 1 Report

This manuscript concluded that the plasma membrane PIP2 selectively modulates the gating of TMEM16A(ac) (slow phase) using the electrophysiological techniques. The electrophysiological techniques and strategies including kinetics analyses are reliable and pharmacological experiments (including PIP2 specificity) are performed carefully. There are a few concerns that need to be addressed.

Major concerns:

1. Authors used HEK-293T mammalian cell expression system in this manuscript and examined the effect of PIP2 depletion on TMEM16A properties. Authors should show the data on the inhibitory effect of Gq-coupled receptor agonist on the TMEM16A(a,c) currents.

2. In ‘Discussion’ section, authors described the disagreement and discrepancy between results by authors and other studies (Page 8, line 258-259 and line 298-300). The readers will need further information to understand the reasons.

3. I strongly recommend that authors should add ‘In conclusions’ in Section 5.

Minor concerns:

1. Page 6, line 182: The word ‘rectification’ is correct.

2. Page 8, line 273, 279, and so on: Ca2+ ion / Iinst current should be amend Ca2+ / Iinst or Ca ions / instantaneous current.

Author Response

Reviewer #1 comments

Major concerns

  1. Authors used HEK-293T mammalian cell expression system in this manuscript and examined the effect of PIP2 depletion on TMEM16A properties. Authors should show the data on the inhibitory effect of Gq-coupled receptor agonist on the TMEM16A(ac) currents.

Response: We thank the reviewer for the constructive comments. We have included our new experimental data regarding the effect of M1 muscarinic receptor (M1R), a Gq-protein coupled receptor (Gq-GPCR) on TMEM16A(ac) current. To perform this experiment, we referred to the protocol of De Jesús-Pérez group (De Jesús-Pérez., 2018). TMEM16A can be activated at voltages > +100 mV in zero intracellular Ca2+ (Xiao et al., 2011). Based on this point, we also measured the whole-cell currents of TMEM16A(ac) at +100 mV in the presence of EGTA with zero Ca2+ in the pipette solution. The TMEM16A(ac) currents were compared before and after activation of M1R with 10 μM Oxo-M in cells expressing TMEM16A(ac) alone or with M1R. Our results show that TMEM16A(ac) currents were inhibited by the activation of M1R with Oxo-M, while no significant reduction observed the currents in cells expressing only TMEM16A(ac), suggesting that TMEM16A(ac) channels can be negatively modulated by Gq-coupled receptors through the depletion of PI(4,5)P2.

We included our new experimental data as Figure 6 in the revised manuscript.

[Images of Figure 6]

Figure 6. Modulation of TMEM16A(ac) channels by M1 muscarinic receptor (M1R). (A) Representative Cl- current traces of TMEM16A(ac) before and after application of the M1R agonist oxotremorine M (Oxo-M). Whole-cell recordings were measured before (t0, black) and 200 s after (t200, gray and red) the application of 10 μM Oxo-M in cells co-transfected without (left) and with (right) M1R. Cells were dialyzed with intracellular solution including 10 mM EGTA and zero Ca2+ and current was measured at +100 mV every 5 s. (B) Normalized mean current ± SEM. TMEM16A(ac) (n = 5), TMEM16A(ac) + M1R (n = 5).

  1. In ‘Discussion’ section, authors described the disagreement and discrepancy between results by authors and other studies (Page 8, lines 258-259 and lines 298-300). The readers will need further information to understand the reasons.

Response: As reviewer’s suggestions, we have included additional information for the lines in the Discussion section (pages #9) of the revised manuscript.

Page 8, line 288-: Ta et al. reported the time constant for recovery to be 7.2 s [39]. This minor disagreement in recovery time may be caused by differences in the duration of applied depolarizing pulses. They conducted the recovery experiment using a double-pulse protocol, where two +100 mV/4 s depolarizing pulses were given with different time intervals. In our study, after applying a single +120 mV/1 s depolarizing pulse, the recovery of TMEM16A from inhibition was measured by applying various interval times. Previous studies showed that the activation of Dr-VSP for less than 1 s with +100-120 mV depolarizing pulses is enough for depleting membrane PI(4,5)P2 [36]. Therefore, because of rapid turnover between the phosphoinositides in cell membrane, the increase in the depolarizing duration would deplete other poly-phosphoinositides, such as PI4P and PI(3,4,5)P3, which might retard the resynthesis of PI(4,5)P2 and current recovery from the inhibition.

Page 9, line 335-345: However, the Tembo et al. [19] and Yu et al. [9] showed that TMEM16A channel activity is slightly enhanced by the application of 100 μM diC8-PI4P and 10 μM diC8-PI(3,4,5)P3, respectively. The discrepancy between those results and our data is likely to have been caused by the accumulation of exogenous phosphoinositides to supramaximal levels in the plasma membrane. Similarly, in the comparative experiments done with receptor stimulation, voltage-sensitive lipid 5-phosphatase, or engineered fusion protein carrying phosphatase, Kruse et al. [40] found that the regulation of KV7 channels by those three methods was inconsistent with other groups who applied exogenous PI(4,5)P2. They speculate that one of the possibilities is due to superphysiological high-concentration of phosphoinositide in cells when exogenous phosphoinositide was applied.

  1. I strongly recommend that authors should add ‘In conclusions’ in Section 5.

Response: We thank the reviewer for this point. We have included the ‘5. Conclusions’ section of revised manuscript as shown below:

  1. Conclusions

This study will expand our understanding of the PI(4,5)P2 regulation of TMEM16A channels in living cells. Using electrophysiological recordings and confocal microscopy, the present works show that two splice variants TMEM16A TMEM16A(a) and TMEM16A(ac) are differentially regulated by membrane lipid PI(4,5)P2. Our results suggest that the slow component of TMEM16A(ac) currents is selectively inhibited by the attenuation of plasma membrane PI(4,5)P2 level and co-expression of PIPKIγ eliminates the inhibition by elevating the PI(4,5)P2 synthesis. In the chemical-induced dimerization assays using exogenous phosphoinositide phosphatases, we confirmed that TMEM16A(ac) current reduction appears only on the dephosphorylation of PI(4,5)P2, but not other phosphoinositides, in a membrane-potential independent manner. Additionally, we observed that the PI(4,5)P2 depletion-mediated current inhibition occurs during Gq-coupled receptor activation. These observations corroborate the role of PI(4,5)P2 as an important modulatory factor of the TMEM16A channel gating.

Minor concerns

  1. Page 6, line 182: The word ‘rectification’ is correct.

Response: We are sorry for the typo. We have corrected the error.

  1. Page 8, line 273, 279, and so on: Ca2+ ion / Iinst current should be amend Ca2+ / Iinst or Ca ions / instantaneous current.

Response: Thank you for your corrections. We have modified those to simply Ca2+ or Iinst in the revised manuscript.

References

  1. De Jesús-Pérez, J.J.; Cruz-Rangel, S.; Espino-Saldaña, Á.E.; Martínez-Torres, A.; Qu, Z.; Hartzell, H.C.; Corral-Fernandez, N.E.; Pérez-Cornejo, P.; Arreola, J. Phosphatidylinositol 4,5-bisphosphate, cholesterol, and fatty acids modulate the calcium-activated chloride channel TMEM16A (ANO1). Biophys. Acta – Mol. Cell Biol. Lipids 2018, 1863, 299–312.
  2. Xiao, Q.; Yu, K.; Perez-Cornejo, P.; Cui, Y.; Arreola, J.; Hartzell, H.C. Voltage- and calcium-dependent gating of TMEM16A/Ano1 chloride channels are physically coupled by the first intracellular loop. Natl. Acad. Sci. USA 2011, 108, 8891–8896.

Reviewer 2 Report

The authors have shown that PI(4,5)P2 specifically regulate the slow Cl- currents conducted by TMEM16A in HEK293T cells, for the TMEM16A(ac) isoform but not the TMEM16A(a) isoform. I recommend publication of the paper.

An important missing piece is, whether PI(4,5)P2 binds directly to the four residues in the c segment or this happens through an allosteric mechanism or if it even involves another protein or lipid component in the HEK293T cells. The authors should comment on this, including if this is not known. Also, it is not clear that these conclusions generated in HEK293T cells would generalize so strongly to other cells with different lipid compositions or whether lipid components other than PI(4,5)P2 may play a role in the slow component of the Cl- current. Since the authors talk about the differential tissue distributions of the a and ac isoforms, this would be another important point for the authors to discuss or present data.

Author Response

Reviewer #2 comments

The authors have shown that PI(4,5)P2 specifically regulate the slow Cl- currents conducted by TMEM16A in HEK293T cells, for the TMEM16A(ac) isoform but not the TMEM16A(a) isoform. I recommend publication of the paper.

An important missing piece is, whether PI(4,5)P2 binds directly to the four residues in the c segment or this happens through an allosteric mechanism or if it even involves another protein or lipid component in the HEK293T cells. The authors should comment on this, including if this is not known.

Response: We thank the reviewer for offering these points. As reviewer’s suggestion, we added a short description about how c-segment regulate PIP2 sensitivity of TMEM16A channels in the Discussion section.

Page 8, line 276-279: Our recent study revealed that c-segment allosterically regulate the interaction of TMEM16A channels with PI(4,5)P2 by altering the structure of PI(4,5)P2-binding sites on the channel protein, while it does not bind to PI(4,5)P2 directly [10].

Also, it is not clear that these conclusions generated in HEK293T cells would generalize so strongly to other cells with different lipid compositions or whether lipid components other than PI(4,5)P2 may play a role in the slow component of the Cl- current. Since the authors talk about the differential tissue distributions of the a and ac isoforms, this would be another important point for the authors to discuss or present data.

Response: We thank the reviewer for pointing out this important query. We described the different possible actions of PIP2 in different tissues in the Discussion section.

Page 10, line 356-366: Of diverse phosphoinositides detected in the inner leaflet of plasma membrane, PI(4,5)P2 is usually the main poly-phosphoinositide (44). PI(4,5)P2 constitutes 0.2-1% of total cellular membrane lipids and 2-5 mol% of total phosphoinositides in the plasma membrane (43, 45). According to this information, PI(4,5)P2 are the second largest phosphoinositides behind PI in the plasma membrane of most mammalian cells. However, the composition of these phosphoinositide is likely to vary depending on cell types and tissues. It was reported that PI(4,5)P2 levels are 5-10 times higher in intact cardiac tissue than isolated myocytes (46). Owing to this reason, the level of PI(4,5)P2 does not change significantly during the activation of muscarinic receptors in intact atrial tissue (47). Therefore, the PI(4,5)P2 effects on TMEM16A may also be different depending on the phosphoinositide compositions of the cells.

Reviewer 3 Report

The authors have studied the Ca2+ activated homodimeric sodium channel TMEM16A,  aka. Anoctamin 1. The focus here is on the influence of PI(4,5)P2 and Ca2+ on the channel function, and, in particular  PI(4,5)P2 binding. As their main result the authors report that depletion of endogenous PI(4,5)P2 inhibits the channel live cells and, importantly, that no phosphoinositides have the same effect. The manuscript is very well and clearly written and, based on my reading, the methods are well chosen and executed. The results shed new light in the functioning of principles of the TMEM16A channels and thus this work provides a meaningful contribution that should be of interest for a broad community. I recommend publication as is.

Author Response

Reviewer #3 comments

The authors have studied the Ca2+ activated homodimeric sodium channel TMEM16A, aka. Anoctamin 1. The focus here is on the influence of PI(4,5)P2 and Ca2+ on the channel function, and, in particular PI(4,5)P2 binding. As their main result the authors report that depletion of endogenous PI(4,5)P2 inhibits the channel live cells and, importantly, that no phosphoinositides have the same effect. The manuscript is very well and clearly written and, based on my reading, the methods are well chosen and executed. The results shed new light in the functioning of principles of the TMEM16A channels and thus this work provides a meaningful contribution that should be of interest for a broad community. I recommend publication as is.

Response: We would like to thank you for your favorable evaluation on our manuscript.

Round 2

Reviewer 1 Report

The authors addressed all comments and revised the manuscript accordingly. I have no more concerns.